# Genetic Mechanisms of Vancomycin Resistance in *Clostridioides difficile*: A Systematic Review

**DOI:** 10.3390/antibiotics11020258

**Published:** 2022-02-16

**Authors:** Taryn A. Eubank, Anne J. Gonzales-Luna, Julian G. Hurdle, Kevin W. Garey

**Affiliations:** 1Department of Pharmacy Practice and Translational Research, University of Houston College of Pharmacy, Houston, TX 77204, USA; taeubank@central.uh.edu (T.A.E.); ajgonz23@central.uh.edu (A.J.G.-L.); 2Center of Infectious and Inflammatory Diseases, Institute of Biosciences and Technology, TX A&M Health Science Center, Houston, TX 77030, USA; jhurdle@tamu.edu

**Keywords:** *Clostridium difficile*, antimicrobial resistance, reduced susceptibility, *van* genes, plasmids, efflux pumps, biofilm

## Abstract

Antimicrobial resistance to treatments for *Clostridioides difficile* infection (CDI) poses a significant threat to global health. *C. difficile* is widely thought to be susceptible to oral vancomycin, which is increasingly the mainstay of CDI treatment. However, clinical labs do not conduct *C. difficile* susceptibility testing, presenting a challenge to detecting the emergence and impact of resistance. In this systematic review, we describe gene determinants and associated clinical and laboratory mechanisms of vancomycin resistance in *C. difficile*, including drug-binding site alterations, efflux pumps, RNA polymerase mutations, and biofilm formation. Additional research is needed to further characterize these mechanisms and understand their clinical impact.

## 1. Introduction

*Clostridioides difficile* infection (CDI) is the most common healthcare-associated infection in the United States with an estimated 223,900 cases in hospitalized patients and 12,800 associated deaths in 2017 [1,2]. For more than four decades, metronidazole and vancomycin have been the leading antimicrobial therapies for CDI, with fidaxomicin also indicated for use. However, metronidazole is no longer recommended as a first-line treatment option. Vancomycin is a glycopeptide antimicrobial that inhibits cell wall synthesis by binding to the D-Ala-D-Ala termini of lipid II, thereby inhibiting peptidoglycan biosynthesis and assembly. Vancomycin has excellent in vitro activity against *C. difficile* and achieves stool concentrations between 500–2000 mg/L [3]. Systemic susceptibility breakpoints vary from concentrations of 2 mg/L (European Committee on Antimicrobial Susceptibility Testing [EUCAST]) to 4 mg/L (Clinical and Laboratory Standards Institute [CLSI]) [4,5]. Clinical labs do not routinely perform *C. difficile* culture and susceptibility testing as a part of the diagnostic work-up, making it difficult to detect resistance. Thus, vancomycin resistance in patients with CDI has not been studied in depth. However, resistance to other antibiotics used to treat CDI, namely metronidazole, has been identified, leading to the question of whether *C. difficile* may also evolve clinically meaningful resistance to vancomycin [6,7,8]. Prescription rates for oral vancomycin increased following the 2017 IDSA/SHEA recommendation of vancomycin as first-line therapy for CDI [9]. This likely also increased the selection pressure for the spread of vancomycin-resistant *C. difficile*.

Vancomycin clinical cure rates have decreased over time [10,11,12,13,14], and range from 82–88%, which is much lower than the cure rates of 93–100% reported in earlier trials [15,16,17,18]. Population surveillance studies have also noted increased *C. difficile* vancomycin minimum inhibitory concentrations (MICs) [19,20,21]. However, the underlying mechanisms of vancomycin resistance in *C. difficile* are still under-characterized, as is their role in treatment failures. In this systematic review, we aim to summarize the body of literature on vancomycin resistance mechanisms in *C. difficile.*

## 2. Results

The literature selection process is summarized in Figure 1. A total of 964 articles were identified for initial screening using the standard systematic review approach, while the increased search efficiency strategy method identified 24 eligible articles. All 24 articles were also identified through the standard approach, with the novel method finding an additional three articles for screening. Upon review of these additional three articles, they did not meet the inclusion criteria. After a full-text review, the same eight articles were identified for inclusion through either literature search strategies [22,23,24,25,26,27,28,29].

Articles were grouped based on the resistance mechanism. For the purposes of this review, ‘resistance’ and ‘reduced susceptibility’ are used interchangeably based on the terminology used in the included literature but is universally defined as a vancomycin MIC of >2 mg/L. Table 1 summarizes the four included articles that specified vancomycin MICs and associated gene(s), while Table 2 summarizes other genes described as indirectly or non-specifically affecting vancomycin resistance.

### 2.1. Binding Site Alterations

#### 2.1.1. Van Genes

The most well-documented mechanisms of vancomycin resistance are target site alterations mediated by *van* genes that modify the terminal D-Ala-D-Ala motif of lipid II, where vancomycin binds to exhibit its mechanism of action. These resistance genes have been well described in *Enterococcus* spp. and are also present in *C. difficile*, albeit without evidence of the same clinical implications [22,23,24,30,31,32,33,34,35,36,37]. Multiple *van* gene clusters, including *vanA, vanB, vanG,* and *vanW*, and *vanZ* orthologs, have been identified in *C. difficile* and have been associated with elevated vancomycin MICs (Table 1) [22,23,24,35]. The expression of these clusters is controlled by a two-component regulatory system, VanSR, that contains VanS, a membrane sensor kinase, and VanR, a cytoplasmic response regulator [23,30]. VanS typically detects the presence of vancomycin, leading to autophosphorylation and transfer of its phosphoryl group to VanR. In turn, phosphorylated VanR binds to the promoter region to induce the transcription of *vanG*. 

Reduced vancomycin susceptibility in *C. difficile* isolates harboring *vanG* has been described in two articles [22,23]. Expression of the *vanG* operon leads to the production of D-Ala-D-Ser rather than D-Ala-D-Ala, altering the vancomycin binding site and decreasing its binding affinity [23,30]. As 85% of *C. difficile* carry a functional *vanG* gene cluster, the mere presence of the gene is not linked with clinical resistance [35,38]. Ramírez-Vargas et al. analyzed 38 isolates of a locally endemic Costa Rican North American pulsed-field variant (NAPCR1) and similarly found all possessed VanG-like sequences but only four isolates were resistant (MIC 4 mg/L) [22]. However, Shen et al. recently demonstrated that levels of *vanG* expression may correlate with resistance in ribotype (RT) 027 after identifying VanSR mutations leading to constitutive *vanG* expression and decreased vancomycin killing [23]. Serial passaging of a reference strain, R20291, was performed to create laboratory mutants demonstrating elevated MICs (8–16 mg/L), which were found to have developed mutations in *vanS*. Eleven clinical isolates with elevated MICs (4–8 mg/L) were then analyzed and found to exhibit similar *vanSR* mutations leading to constitutive *vanG* expression (Table 1).

**Table 2 antibiotics-11-00258-t002:** Genes indirectly/non-specifically associated with vancomycin resistance.

Ref.	Gene	Genetic Mechanism (If Described)	Encoded Protein	Function	Proposed Mechanism of Resistance
[29]	*agrD_1_*	Overexpression	Accessory gene regulator D1 (AgrD_1_)	Quorum sensing generation/modulation	Unknown
[27]	*cd2068*	Upregulation	CD2068	ABC transporter pump	Drug efflux*
[25]	*cd3659*	Mutation (Gly982Thr)	DNA exonuclease/phosphodiesterase	--	Unknown
[28]	*cdtA*	Presence **	Toxin A (TcdA)	Enterotoxin	No impact
[28]	*cdtB*	Presence **	Toxin B (TcdB)	Cytotoxin	No impact
[28]	*cwlD*	Presence **	N-acetylmuramoyl-L-alanine amidase	Spore germination	No impact
[24,28,29]	*cwp84*	Overexpression	Cysteine protease (Cwp84)	Surface layer protein maturation/adhesion/biofilm regulation	Biofilm production
[28]	*fliC*	Presence **	Flagellin (FliC)	Adhesion/biofilm regulation	Biofilm production
[28,29]	*luxS*	Presence **	LuxS	Autoinducer-2 synthesis/quorum sensing modulation/biofilm regulation	Biofilm production
[26]	pX18-498_006 ***	Presence	N-acetylmuramoyl-L-alanine amidase	Cell wall integrity	Unknown
[25]	*sdaB/cd3222*	Mutation (Ala295 deletion)	L-serine deaminase	--	Unknown
[29]	*sigH*	Overexpression	Sigma factor (SigH)	Sporulation regulation	Biofilm production
[28]	*sleC*	Presence **	SleC	Spore germination	Biofilm production
[29]	*slpA*	Presence	S-layer protein (SlpA)	Surface layer composition/ adhesion/biofilm regulation	No impact
[28,29]	*spo0A*	Overexpression	Spo0A	Sporulation/biofilm regulation	Biofilm production

* Effective vancomycin efflux following introduction in *E. coli* but no difference in vancomycin efflux observed when introduced to *C. difficile*. ** Proposed effects and mechanism based on creation of dysfunctional mutant strains as described by Ðapa et al. [28]. *** Plasmid encoding multiple genes hypothesized to impact vancomycin susceptibility by Pu et al.; protein listed is most likely to be related as per author [26].

Although the process by which resistance is conferred is less well defined, *vanA, vanB, vanW*, and *vanZ* have been associated with vancomycin resistance in *C. difficile* as well [24]. These *van* genes disseminate vancomycin resistance amongst various bacterial genera via plasmid acquisition, with VanB-type spread specifically linked to plasmid Tn1549 [26,33,39]. Saldanha et al. submitted seven clinical isolates from Brazil for whole-genome sequencing to identify and better understand the relationship between vancomycin-resistance genes and MIC in *C. difficile* [24]. Five out of the seven isolates had elevated MICs and one or more *van* genes present (Table 1). However, the two isolates that were susceptible to vancomycin also contained *vanW* and *vanZ* genes, supporting the theory that the presence of *van* genes alone does not correlate with vancomycin resistance. Future studies investigating the role of specific genetic mutations and/or gene expression levels are needed to provide a framework for interpreting the implications of these other *van* genes.

#### 2.1.2. MurG 

MurG is a glycosyltransferase enzyme responsible for converting lipid I to lipid II in peptidoglycan biosynthesis, which results in the formation of the cell wall [40]. Alterations in this pathway may affect vancomycin activity since vancomycin binds the D-Ala-D-Ala terminal on lipid II to prevent cell wall formation. Leeds et al. conducted experimental evolutions and determined that a mutation in the *murG/cd2725* gene was one of three mutations identified in a mutant with elevated vancomycin MICs [25]. This mutation caused a conserved proline to leucine change (P108L) in the MurG enzyme near the second of three G loops needed for binding the phosphate groups of UDP-GlcNAc. As the isolate exhibiting this change also had two additional mutations present, it is unknown what specific role this *murG* mutation played in decreasing vancomycin binding and/or activity.

### 2.2. Plasmid Acquisition of N-Acetylmuramoyl-L-Alanine

A plasmid is typically a small circular DNA strand that can replicate independently in microorganisms. Plasmids can be transferred between bacteria through horizontal gene transfer, thus providing genes of antimicrobial resistance. The gut microbiome is an abundant source of extrachromosomal elements to be potentially transferred amongst bacteria due to the diverse inhabitants. Pu et al. identified the plasmid pX18-498 in CDI-positive patients who were non-responders to vancomycin treatment [26]. When this plasmid was transferred to *C. difficile* isolates from vancomycin responders (MIC < 1 mg/L), the isolates exhibited an 8-fold decrease in susceptibility to vancomycin. The authors identified the gene encoding N-acetylmuramoyl-L-alanine amidase as one of the top five differentially expressed genes contained within the plasmid pX18-498. N-acetylmuramoyl-L-alanine amidase is essential for susceptibility to antibiotics that target the cell wall in other bacterial species, and implantation of pX18-498 with this amidase-encoding gene into *C. difficile* caused cell rupture and decreased permeability. Furthermore, mice infected with *C. difficile* strains carrying pX18-498 demonstrated a more severe disease phenotype. Studies are needed to confirm this gene expression is indeed the cause of the observed decreased vancomycin susceptibility and to further elucidate possible resistance mechanisms *C. difficile* can acquire from plasmid acquisition.

### 2.3. Efflux Pumps

Efflux pumps are active transporters residing in the bacterial cell membrane that utilize an energy source to transport solutes in or out of the cell. One major family of efflux pumps is the ATP-binding cassette (ABC) transporters, which use adenosine triphosphate (ATP) to transport solutes ranging in size from simple ions to larger molecules, such as antibiotics [27,41]. These transporters are known to be a major cause of multidrug resistance in many bacteria and have been identified in several Clostridium species as well [42,43]. In *C. difficile*, cationic antimicrobial peptides (CAMPs), including vancomycin, have been shown to induce the expression of a proposed ABC transporter operon, which in turn decreased the effectiveness of various CAMP antibiotics [44]. However, specific ABC transporters had not been recognized as contributing to multidrug resistance in *C. difficile* until Ngernsombat et al. identified and characterized the CD2068 transporter [27]. 

CD2068 was identified when genome analysis of a reference CD630 strain identified it as having a high level of homology to two other known ABC transporters in *C. hathewayi* and *C. perfringens* [27,42,43]. Following identification, Ngernsombat et al. conducted various experiments to determine the function of CD2068 [27]. First, the gene expression of *cd2068* was shown to significantly increase following exposure to various antibiotics, including vancomycin (0.25 mg/L). When cloned and functionally characterized in *Escherichia coli*, CD2068 increased the half maximal inhibitory concentration (IC50) of vancomycin and was associated with 2.6 times higher relative resistance. However, when a *cd2068* gene, following insertional activation and complementation, was introduced into *C. difficile*, no significant difference in vancomycin IC50 was observed. The reason(s) behind the lessened ability of CD2068 to cause multidrug resistance in *C. difficile* vs. *E. coli* is not known, but the authors offer several hypotheses including poor natural expression of CD2068 in *C. difficile*, compensation by other ABC transporters in their mutant strain, and/or the presence of other mechanisms mediating resistance to certain antibiotics in *C. difficile*. Regardless, 243 other genes in the CD630 genome are thought to encode for putative ABC transporters, offering a promising line of research in this area [45]. Clearly, more studies are needed to identify other efflux pumps and their role in contributing to *C. difficile* vancomycin resistance.

### 2.4. RNA Polymerase Mutations

Alterations in the genes encoding bacterial RNA polymerases have been well described for antibiotics that inhibit RNA polymerase as a mechanism of action, such as mutations in *C. difficile rpoB* or *rpoC* leading to rifampicin or fidaxomicin resistance, respectively [46,47,48]. However, *rpoB* mutations have also been associated with vancomycin and daptomycin resistance in *Staphylococcus aureus*, despite these drugs not targeting RNA polymerase [49,50]. Conversely, *rpoB* mutations have not been described as contributing to vancomycin resistance in *C. difficile*. Instead, Leeds et al. described a mutation in *rpoC/cd0067* in an experimentally evolved mutant with a vancomycin MIC of 16 mg/L [25]. The mutation was a single nucleotide change involving a D244Y substitution in the β’ subunit of RNA polymerase. The authors postulated that *rpoC* mutations lead to changes in global gene expression that may affect multiple pathways, therefore impacting vancomycin activity, but further research is needed to investigate the exact mechanism of vancomycin resistance resulting from this mutation.

### 2.5. Biofilm Production

A biofilm is a structure of aggregated microbial communities displaying reduced metabolic activity [51]. Biofilms can serve as a physical barrier preventing host immune responses and sufficient concentrations of antimicrobials from reaching the site of infection, and have accordingly been associated with antimicrobial resistance and recurrent infections [52]. Additionally, the slowed metabolic state of the microbial species can render antibiotics, whose mechanism of resistance relies on fast dividing cells, less active [51,53]. Although the ability of *C. difficile* and related Clostridium species to form biofilms in the intestine has been described and linked to antimicrobial resistance, few articles describing the genetic mechanism(s) behind biofilm development and antimicrobial resistance have been published [28,54,55]. 

Two studies have implicated *C. difficile* biofilm formation with decreased vancomycin susceptibility [28,29]. Both conducted in-depth analyses to determine molecular and genetic factors contributing to their findings as discussed by functional category. The first was conducted by Ðapa et al. who developed a biofilm assay and used it to measure the growth of two *C. difficile* strains (CD630 and R20291) [28]. Growth was measured in 1-, 3-, and 5-day-old biofilms following exposure to 20 mg/L of vancomycin (100 times greater than the MIC). The *C. difficile* survival percentage was 5- and 12-fold higher in the 1- and 3-day old biofilms, respectively, than in planktonic *C. difficile* without a biofilm, demonstrating potentially decreased vancomycin efficacy. The second was performed by Tijerina-Rodriguez et al. who analyzed antimicrobial susceptibilities in biofilm- and non-biofilm-producing clinical isolates from Mexico [29]. Vancomycin MICs were up to 100 times higher in biofilm-producing cells when compared to planktonic cell counterparts. Additionally, a sub-group analysis comparing isolates from patients with recurrent CDI (R-CDI) and nonrecurrent CDI (NR-CDI) demonstrated that isolates from R-CDI patients had significantly higher rates of reduced vancomycin susceptibility (28%) than those from NR-CDI patients (9%; *p* = 0.013). Genetic investigations in both studies found the development of a mature biofilm to be multifactorial, as summarized in Table 2 and detailed below. 

#### 2.5.1. Surface Factors Involved in Biofilm Formation

Various factors within the surface layer of *C. difficile*, such as cell wall proteins (CWP) and adhesins (flagella or pili), are thought to play important roles in biofilm formation and early bacterial adhesion [28]. Specifically, Cwp84 is a surface-associated cysteine protease that has been shown to contribute to surface layer protein maturation and aid in degrading host tissues [56,57]. Ðapa et al. seemed to confirm this finding through a demonstration that mutants with a deletion of *cwp84* had a dramatic decrease in biofilm formation, with larger defects observed in early biofilm growth on day 1 compared to days 3 and 5 [28]. Tijerina-Rodriguez et al. measured and linked *cwp84* expression levels to worse clinical outcomes after observing isolates from patients with R-CDI had significantly higher *cwp84* expression than isolates from patients with NR-CDI (4.31 vs. 0.29 relative mRNA expression; *p* = 0.01) [29]. *cwp84* was also identified in three of the five vancomycin-resistant isolates submitted for whole-genome sequencing in the study conducted by Saldanha et al. but was additionally present in one of the two non-resistant isolates [24].

Similarly, flagella are known to contribute to mature biofilm formation in other bacterial species, but with a lesser understood function in clostridial species [58]. Ðapa et al. mutated the fliC gene encoding flagellin, a major component of flagella, in *C. difficile,* and demonstrated a significant decrease in biofilm production [28]. In contrast to the growth changes seen with *cwp84* mutants, the decrease in formation was observed in day 5 biofilms, but not on days 1 or 3, indicating flagella may be more important for the later stages of biofilm formation.

#### 2.5.2. Sporulation and Germination 

*C. difficile* sporulation is initiated under stress and is mainly regulated by the Spo0A transcription factor. However, Spo0A is thought to switch between controlling sporulation and biofilm growth depending on the concentration of phosphorylated Spo0A, a process well described in Bacillus species [59,60]. As *spo0A* is highly conserved between Bacillus and Clostridium species, Ðapa et al. studied the relationship between *spo0A* and biofilms in *C. difficile* [28]. A *spo0A* mutant with decreased sporulation was created and found to exhibit significantly less biofilm growth than strains with wild-type *spo0A*. Tijerina-Rodriguez et al. also looked at differences in the gene expression of *spo0A* and *sigH*, another sporulation regulator encoding the sigma factor of sporulation, in their clinical cohorts [29]. *sigH* and *spo0A* expression were significantly higher in biofilm-producing R-CDI isolates than NR-CDI isolates (relative *sigH* mRNA expression 6.91 vs. 0.57 and relative *spo0A* mRNA expression 47.40 vs. 0.31, respectively; *p* < 0.01). As previously discussed, both studies described decreasing vancomycin susceptibility associated with biofilm presence, and thus *sigH* and *spo0A* appear to play potential roles in mediating this form of vancomycin resistance [28,29].

#### 2.5.3. Quorum Sensing 

Quorum sensing is a density-dependent form of cell-to-cell signaling used by microbial communities to coordinate various cellular processes, including the development of antimicrobial resistance [61]. LuxS is one of many regulatory molecules involved in quorum sensing in *C. difficile* and has been shown to both mediate communication within biofilms as well as contribute to biofilm formation itself [28,62,63]. Following the groups’ findings that biofilms appeared to decrease vancomycin efficacy, Ðapa et al. created a *luxS* mutant and observed a dramatically decreased ability to form biofilm [28]. Tijerina-Rodriguez et al. measured the expression of *luxS* in addition to *agrD1*, which is part of the Agr quorum sensing system homologous to the Agr system in *Staphylococcus aureus*; in *S. aureus*, Agr is known to influence biofilm formation and AgrD is the precursor of the autoinducing peptide (quorum sensing signal) [29,64]. Interestingly, the group found a significantly higher expression of *agrD1* in biofilm-producing R-CDI isolates vs. NR-CDI isolates (relative mRNA expression 14.02 vs. 0.38; *p* < 0.01) but no difference in *luxS* expression. Although this does not diminish the possibility of *luxS* playing a role in biofilm production, further exploration of *agrD1* is warranted to explore its exact role in quorum sensing and antimicrobial resistance [29].

Although biofilms have been shown to be associated with increased vancomycin MIC and recurrent CDI, biofilm formation is multifactorial, and current research has been unable to definitively associate any one mechanism specifically with vancomycin resistance.

## 3. Materials and Methods

A systematic literature search was conducted using two separate strategies. A PubMed search was conducted using the keywords (“*Clostridium difficile*” OR “*Clostridioides difficile*”) AND (“resistance”) AND (“vancomycin” OR ‘teicoplanin”). A filter for the English language was applied. Each article from the search was evaluated for inclusion regardless of publication date. Included studies must have been conducted in *C. difficile* and included a measure of vancomycin susceptibility and description of the underlying genes associated with susceptibility changes. Articles categorized as reviews, commentaries, opinions, meta-analyses, and systematic reviews were excluded. The references of the included publications were evaluated for relevant articles and cross-referenced with previous related reviews.

A second literature search method designed for increased search efficiency was conducted in parallel. A PubMed search was performed using the keywords: (“*Clostridium difficile*” OR “*Clostridioides difficile*”) AND (“resistance”) AND (“vancomycin” OR ‘teicoplanin”). Unlike the previous search strategy, not all results were screened, and instead, select journal articles discussing the role of specific genes on vancomycin susceptibilities in *C. difficile* were chosen for screening. Then, articles designated as ‘Similar articles in PubMed’ were evaluated and used to compile a list of articles that subsequently underwent title and abstract review using the same inclusion criteria as above. The references of the included publications from this search strategy were evaluated for relevant articles and were cross-referenced with previous related reviews.

## 4. Conclusions

As few antibiotic treatment options are available for CDI, an increasing amount of selection pressure is placed on the development of antibiotic resistance. Oral vancomycin is increasingly used as the first-line drug due to recent treatment guideline updates, costs and availability concerns [8,9,14,65]. However, clinical resistance to vancomycin is yet to be appreciated as a major threat in CDI as separating the effects of host factors, antibiotic failure, and infection recurrence in reported poor outcomes remains a challenge. Future studies will need to better understand the role of other microbiota in the gut to elucidate resistance in vancomycin. In this review, we aimed to summarize the existing literature on genes associated with vancomycin resistance to serve as a primer for clinicians and researchers working in this area. As the clinical significance of these resistance mechanisms become clearer, methods to identify these gene targets in clinical practice will become necessary.

## Figures and Tables

**Figure 1 antibiotics-11-00258-f001:**
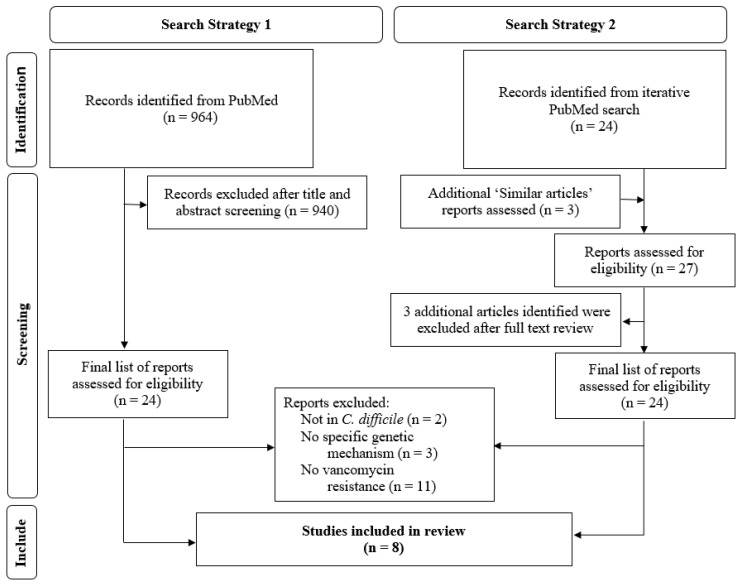
PRISMA chart of literature flow with a comparison of traditional and novel literature search process. http://www.prisma-statement.org/ (accessed on 13 January 2022).

**Table 1 antibiotics-11-00258-t001:** Summary of genes associated with elevated vancomycin MICs.

Ref.	No. Isolates ^†^	Strain Origin	Ribotype	ST Type	Other	Gene (Mutation, If Known)							VAN MIC (mg/L)
						*vanG/vanSR*	*vanA*	*vanB*	*vanW*	*vanZ*	*murG*	*rpoC*	
[22]	4	Clinical	012	54	NAP_CR1_	*vanG*							4
[23] *	1	Laboratory	027	1	WS2/ R20291	*vanS* (Arg314Leu)							8
	1	Laboratory	027	1	WS4/ R20291	*vanS* (Gly319Asp)							16
	1	Clinical	027	1	MT1470	*vanS* (Ser313Phe)							8
	1	Clinical	027	1	MT5006	*vanS* (Thr349Ile)							8
	9	Clinical	027	1	see note ^‡^	*vanR* (Thr115Ala)							4–8 ^‡^
[24]	1	Clinical		35			*vanA*		*vanW*				4
	1	Clinical	014-020	2				*vanB*					8
	1	Clinical		42				*vanB*					4
	1	Clinical		67				*vanB*	*vanW*				>16
	1	Clinical	012	54					*vanW*	*vanZ*			4
[25] *	1	Laboratory	087		NB95013/ATCC 43255						*murG/cd2725* (Pro108Leu)		16
	1	Laboratory			NB95026							*rpoC/cd0067* (Gly733Thr)	16

Abbv: Ref., reference; no., number; ST, multilocus sequence type; MIC, minimum inhibitory concentration; VAN, vancomycin. Notes:* These studies utilized serial passaging and selection for mutants with decreased susceptibility. † Number of isolates expressing resistance, defined as vancomycin MIC > 2 mg/L [4,5]. ‡ 7 isolates from the Texas Medical Center had MICs = 4 mg/L and 2 isolates from Israel had MICs = 8 mg/L

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
