# Peer review of "Genetic Mechanisms of Vancomycin Resistance in Clostridioides difficile: A Systematic Review"

_antibiotics, 2022, doi:10.3390/antibiotics11020258_

Round 1
Reviewer 1 Report
Interesting review paper on vancomycin resistance in C. difficile.
The introduction could be enlarged by emphasizing the role of the normal gut microbiota. C.d emerges when the normal mainly anaerobic gut mirobes are destroyed by antibiotics. Metronidazole is not a suitable agent as nearly all anaerobes naturally found in the gut are susceptible (which will not help to restore the normal microbiota). Vancomycin is not effective agains many indigenous bacteria.
The frequency of vancomycin resistance should also be mentionned.
In view to open fields to other types of treatment in case of vancomycin resistance, the discussion should mention fidaxomycin, fecal transplantation....
Minor remark: pay attention to the use of italics in Latin names of bacteria. As it is always the case for C.d, it is not the case for Bacillus and Clostrium line 199, 247 and 248. Check throughout the manuscript!
Author Response
Interesting review paper on vancomycin resistance in C. difficile.
**Thank you for your review and helpful comments.
The introduction could be enlarged by emphasizing the role of the normal gut microbiota. C.d emerges when the normal mainly anaerobic gut mirobes are destroyed by antibiotics. Metronidazole is not a suitable agent as nearly all anaerobes naturally found in the gut are susceptible (which will not help to restore the normal microbiota). Vancomycin is not effective agains many indigenous bacteria.
**This is a fabulous future area of focus. Have included this in the conclusion section
The frequency of vancomycin resistance should also be mentionned.
**It is a great point but there has not been a good study to understand the incidence or prevalence of vanco resistance in C diff. This is an active, ongoing project in our lab
In view to open fields to other types of treatment in case of vancomycin resistance, the discussion should mention fidaxomycin, fecal transplantation....
**Added fidaxomicin to the introduction
Minor remark: pay attention to the use of italics in Latin names of bacteria. As it is always the case for C.d, it is not the case for Bacillus and Clostrium line 199, 247 and 248. Check throughout the manuscript!
**Good point, thank you.
Reviewer 2 Report
Dear authors
The present review meets the global problem of antimicrobial resistance due to treatments. The work is focused on Clostridioides and is very well structured form. It summarized scientific data that will be beneficial to address future research in this field.
Author Response
Thank you for the review of our paper and your comments
Reviewer 3 Report
In this review, the authors have described gene determinants and mechanisms of vancomycin resistance in C. difficile, including drug binding site, efflux pumps, RNA polymerase mutations, and biofilm formation. This review paper is comprehensive and informative. The paper can be accepted in present form.
Author Response
Thank you for the review and your comments on our paper